# Microwave Absorption Properties of Fe$_3$O$_4$ Particles Coated with Al via Rotating Magnetic Field Method

**Ivan Shorstkii** *  **and Maxim Sosnin**

Advanced Technologies and New Materials Laboratory, Kuban State Technological University, 350072 Krasnodar, Russia; maksim-sosnin7@mail.ru
* Correspondence: i-shorstky@mail.ru

**Abstract:** Soft magnetic composites (SMCs) of Fe$_3$O$_4$ particles coated with Al nanoparticles were prepared using the rotating magnetic field method, and the microwave absorption properties and microstructures of these composites were investigated. The results show that a well-distributed Al nanoparticles coating layer was formed on the surface of the Fe$_3$O$_4$ particles upon mechanical friction and rotating magnetic field distribution. Scanning electron microscopy and X-ray diffraction XRD studies show that the rotating magnetic field method can produce a uniform coating of the aluminium layer on the Fe$_3$O$_4$ particles. Compared with common composites from Fe$_3$O$_4$ particles, SMCs of Fe$_3$O$_4$(Al) particles have stronger magnetic loss behaviour and weaker dielectric loss ability, as well as good reflection characteristics over a wide frequency range. The minimum reflection loss (*RL*) is −16.2 dB at 12.0 GHz for a corresponding thickness of 5 mm obtained for SMCs of Fe$_3$O$_4$(Al) particles. The presented rotating magnetic field method used in the Fe$_3$O$_4$ particles coating process with Al nanoparticles has great potential in composite materials synthesis with different morphology and areas of application.

**Keywords:** composite material; coating; rotating magnetic field; cladding; microwave absorption

## 1. Introduction

Soft magnetic composites (SMCs), manufactured via powder metallurgy processing, are of significant interest in the field of core materials [1]. Electromagnetic (EM) interference pollution, arising from the rapidly expanding business of communication devices, such as mobile telephones, local area network systems and radar systems, has attracted great interest to scientists in the synthesis of new types of microwave absorption materials [2]. Among the candidates for EM wave absorbers, core-shell-type composite materials may be considered as potential candidates for microwave absorption in a wide frequency range [3–6]. In such composite materials, it is possible to combine strong absorption, low density and high resistivity properties. At the same time, it is important to ensure a uniform morphology of particles in the matrix with the achievement of a strong interfacial connection between the matrix and reinforcement [7].

Soft metallic magnets are usually considered the core for such materials. However, for example, the well-known magnetite Fe, despite its magnetic properties, has a limited absorption frequency range, which prevents its wide practical application. The use of Fe$_3$O$_4$ core-shell particles, in combination with other materials–metals and materials–dielectrics, as a shell causes new synergistic effects in the form of improved EM absorption and reflection characteristics. A wide range of core shells in the form of polymer materials [8,9], carbon composites [7,10], or aluminium particles [11] has been studied for Fe particles.

Depending on the properties of shell material, a hydrothermal deposition method [10], mechanical grinding [11], polymerization [12], or a combination of several methods [13,14] are mainly using for the core coating process. The core-shell connection leads to the appearance of a skin effect caused by strong eddy currents [15]. The highest current density can be achieved at the surface and decreases with penetration into the conductor. For

a ferromagnetic conductor, the skin effect leads to a partial weakening of the internal magnetic field and, as a result, a deterioration in the complex permeability [15]. Our study addressed a research question: does a strong core-shell bond is necessary for SMC with strong EM absorption characteristics?

In this study, we designed a novel porous core-shell composite with dense particles packaging as a new type of microwave absorption materials using the rotational magnetic field method.

In comparison with a static magnetic field (SMF) [16], strong magnetic field [17] and pulsed magneto-oscillation (PMO) [18] suggested method has few advantages: (1) an ability to control particles packaging density (from loose to dense structure); (2) possibility to provide core-coating process using the dynamic magnetic platform.

At the same time, the rotating magnetic field can be used as a method for a $Fe_3O_4$ microparticles coating process by Al nanoparticles. In addition, for obtained composite materials the hypothesis of the skin effect influence on EM absorption characteristics is considered.

## 2. Materials and Methods

Microspherical $Fe_3O_4$ particles with an average size of $60 \pm 10$ μm were used in the current study. Aluminium nanoparticles with an average size of $60 \pm 10$ nm were used as the shell for microspherical $Fe_3O_4$ particles. Paraffin was used as a matrix for composite materials. All materials were purchased from a local powder metallurgy company (Riceshell Inc., Krasnodar, Russia).

### 2.1. Preparation of Porous Fe₃O₄/Al Composite

$Fe_3O_4$(Al) composite materials were prepared by the rotating magnetic field (RMF) method [19,20]. Briefly, a $Fe_3O_4$ SMC synthesis consists of two steps (Figure 1). First, microspherical $Fe_3O_4$ particles were placed in a glass tube. A rotating magnetic field of permanent magnets with the orientation N–N or S–S was created around the glass tube by rotating permanent magnets in a nozzle (Figure 2).

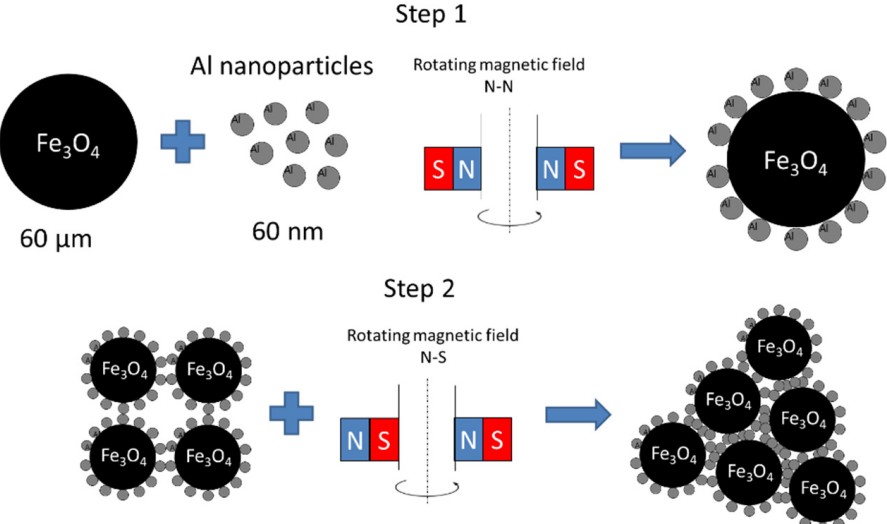

**Figure 1.** Scheme of $Fe_3O_4$ particles coating by Al nanoparticles using N–N rotating magnetic field (step 1) and composite material synthesis using N-S rotating magnetic field (step 2).

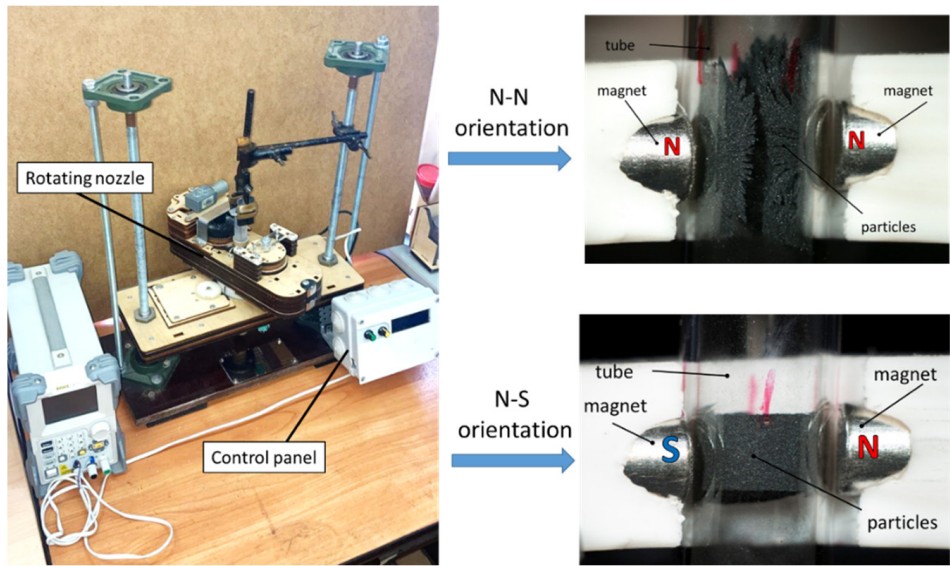

**Figure 2.** RMF setup with different permanents magnets orientation and particle structure.

The rotating magnetic field (RMF) method operated using patented technology and shown in Figure 2. Visualization process of RMF method application is shown on the Video S1 and S2 (Supplementary Materials). The glass tube was fixed to exclude scrolling. During magnetic field rotation Al nanoparticles were added to the $Fe_3O_4$ particles in a mass ratio of 1:9. Due to the N–N orientation of the permanent magnets, the composite mixture of $Fe_3O_4$ and Al particles was uniformly mixed. Mechanical friction of the microspherical $Fe_3O_4$ particles and Al nanoparticles cladding process ensured. Permanent magnets rotation process was carried out until the visible homogeneity of the mixture.

At the second step, permanent magnet polarity was changed to N-S and the paraffin matrix was added to the $Fe_3O_4(Al)$ mixture in a volume ratio of 1:1. This composition was heated up to the paraffin melting point using an alcohol burner. After achieving the necessary uniformity, the rotation of the permanent magnets stopped and the mixture was allowed to cool to ambient temperature. Finally, the product was cut into $23 \times 10$ mm$^2$ samples with 5 mm thicknesses (marked in FAC). During the cutting process, the condition of magnetic field direction along the long side of the samples was followed. For a comparison process of absorption characteristics, a composite material from $Fe_3O_4$ microspherical particles without Al was obtained by the same method (marked in FC).

*2.2. Characterizations*

Phase analysis was tested by the powder X-ray diffraction (XRD) patterns (Shimadzu XRD–7000 S diffractometer, Shimadzu, Kyoto, Japan) using Cu K$\alpha$ radiation ($\lambda$ = 1541 Å) with 40 kV scanning voltage, 30 mA scanning current.

A JSM-6360 and JSM 7500 (Jeol, Tokyo, Japan) scanning electron microscopy (operated at an acceleration voltage of 10.0 kV and 15.0 kV and equipped with an energy dispersive X-ray spectroscopy) were used to observe the morphology features and sizes.

The elemental composition of the composite material was studied using an EDX-8000 (Shimadzu, Kyoto, Japan) X-ray fluorescence spectrometer. For that, EDX-8000 spectrometer operated at 15 to 50 kV for light (Na, Mg, K) and other elements (V, Cr, Fe, Rb, Cs) with software quantitative analysis.

In the simplest case, an electromagnetic wave initially travelling in the free space hits a barrier. A vector network analyser (VNA) measures the phase and magnitude of the scattering parameters (S-parameters). The S-parameters allow obtaining simultaneously the reflected and transmitted power over a given frequency range. In a two port VNA there are four S-parameters S11, S12, S21, and S22. The S$_{[ij]}$ parameter is the fraction of signal reflected back to the same port where the signal was initially injected when $i = j$

($S_{11}$ and $S_{22}$), and the fraction of signal transmitted from Port$_{[i]}$ to Port$_{[j]}$ through the material under test when i $\neq$ j ($S_{12}$ and $S_{21}$). In our work, S parameters were tested by Keysight ENA E5080B vector network analyser (Keysight, Santa Rosa, CA, USA) using the transmission line method. Rectangle-shaped samples ($23 \times 10$ mm$^2$) and waveguide VP1 $-23 \times 10 \times 100$ mm$^3$ with self-made flange used for measurements. Afterwards, software which had been installed in Keysight ENA calculated the $\varepsilon'$, $\varepsilon''$, $\mu'$, and $\mu''$ values. Finally, the RL value was calculated using the method from the literature [21,22].

## 3. Results and Discussion

Derived from the XRD pattern analysis of $Fe_3O_4$ particles, it is found in Figure 3 that these strong diffraction peaks at 30.1°, 35.40°, 37.2°, 43.0°, 53.5°, 56.9°, 62.6° and 74.0° can be can typically be indexed to the spinel phase of $Fe_3O_4$ (220), (311), (222), (400), (422), (511), (440) and (533) in accordance with the standard data of JCPDS Magnetite No. 00-019-0629.

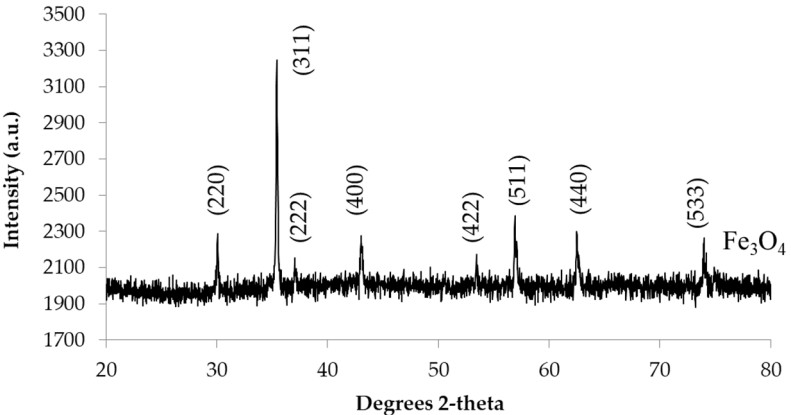

**Figure 3.** X-ray image of the studied $Fe_3O_4$ particles.

The SEM-image of the FC sample is shown in Figure 4. It can be seen that the size of $Fe_3O_4$ particles is 60−70 µm. The morphology feature of the FC sample is shown in Figure 4b. It can be seen that the FC sample presents porous $Fe_3O_4$ structures with dense particles packing. The formed array of $Fe_3O_4$ particles, in accordance with rotating magnetic field action [20], has a porous channel structure with pores with a diameter of approximately 15 µm.

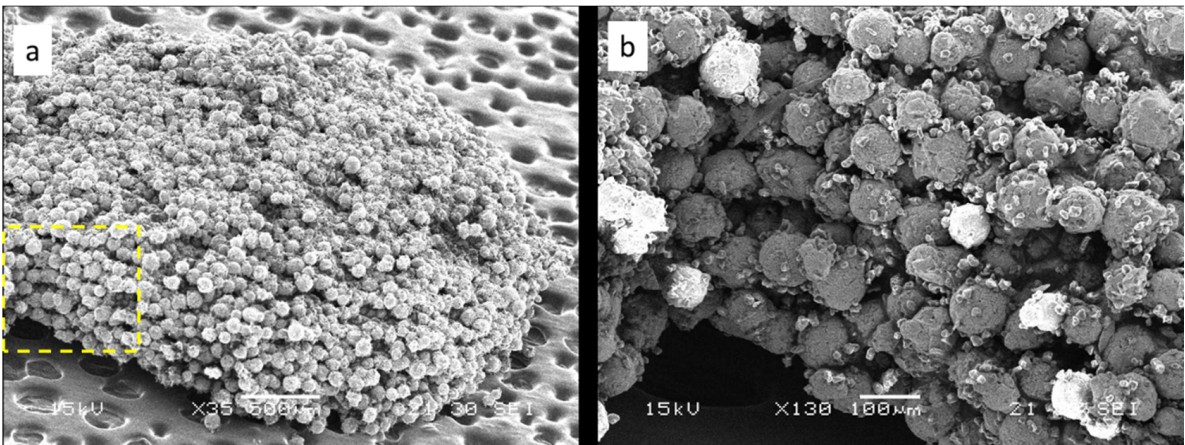

**Figure 4.** SEM image of $Fe_3O_4$ microspherical particles array of the FC sample after application of rotating magnetic field method (**a**) and the surface of $Fe_3O_4$ particles at 40,000× magnification (**b**).

FAC sample morphology at the fractures is shown in Figure 5. Further magnifying the single particle reveals that the surface of $Fe_3O_4$(Al) particle is covered by numerous smooth spherical Al nanoparticles structures (Figure 5b,c, white surface). The selected region in Figure 5b magnified to identify the Al shell and shown in Figure 5c. The EDX results show the presence of Al in FAC sample in 11:89 mass ratio, which corresponds well to the thin layer of shell particles. From a mechanical process point of view, the cladding of particles was carried out by applying mechanical friction force between the particles from the rotational magnetic field with N–N magnets orientation (Figure 2). $Fe_3O_4$ particles were initially oriented along the magnetic field lines and, during the superposition of the rotational magnetic field, had two axes of rotation: one in the centre of the array, aligned with the axis of the glass tube, the other perpendicular to it and passing through the centre of mass of the array of particles. Due to this complex motion, the Al nanoparticles were deposited evenly over the entire surface on the surface structure of the $Fe_3O_4$ particles (Figure 5b). The aluminium shell highlighted in white colour on the SEM-image (Figure 5b).

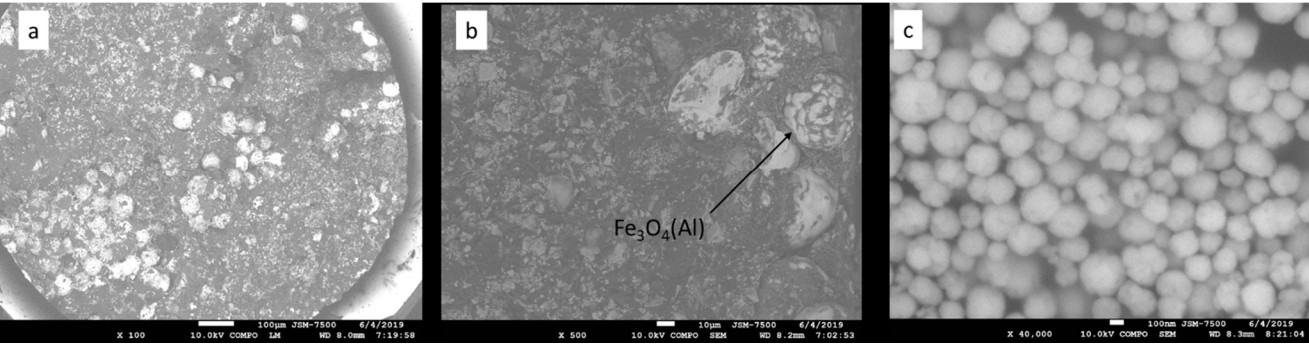

**Figure 5.** SEM images of the composite material FAC in a paraffin matrix with $Fe_3O_4$ particles covered by Al nanoparticles at × 100 magnification (**a**) at 500× magnification (**b**) and the surface of the $Fe_3O_4$(Al) particle at 40,000× magnification (**c**).

Changes in magnetic flux passing the spherical particles array create eddy electric currents [23,24]. The magnetic field of these eddy currents coincides with the direction of the external magnetic field $B_{ext}$. According to the fact, that the thickness of the aluminium shell is quite smaller than the size of the $Fe_3O_4$ particle, the hypothesis put forward in the current paper is relevant. Our hypothesis based on $Fe_3O_4$ particles magnetization changes due to the surface charges appearance in the aluminium shell. In such a system, the circulation currents appear on the core-shell surface. According to the $Fe_3O_4$(Al) SEM-images we put the assumption that there is a continuous aluminium layer on the surface of $Fe_3O_4$ particles.

It is well known that, for a spherical particle, the maximum eddy current appears along the equator [24]. We construct a model of a $Fe_3O_4$ microparticle with radius *a*, covered by Al nanoparticles with thickness *d*. Herewith, the external electromagnetic field the magnetization vector of the core ($Fe_3O_4$ particle) is parallel to the vector of the external magnetic field $B_{ext}$. The vector of voltage is perpendicular to the external electromagnetic field the magnetization vector (Figure 6).

The magnetic moment of the circulation currents of a spherical $Fe_3O_4$(Al) placed in a magnetic field can be determined by the expression [23,24]:

$$m = \int_0^{\pi/2} \left(2M\pi(a+d)^3\right) \sin^3 \theta \, d\theta \tag{1}$$

where *M* is the magnetization of the spherical particle from the external magnetic field, Gs; *a* is the radius of the spherical particle, m; *d* is the thickness of the nanoparticle layer Al, m; $\theta$ is the angle between the magnetization vector and the plane of the circle. Under the condition of $\theta = 90°$ ($\sin\theta = 1$), the magnetic moment of the eddy currents will be calculated as $m_{FA} = \frac{4\pi}{3} \cdot (a+d)^3 M$.

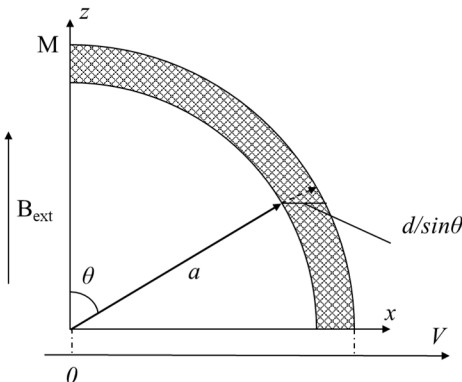

**Figure 6.** $Fe_3O_4(Al)$ single particle model.

The magnetic moment of the eddy current can be expressed in terms of the electric potential [23,24]:

$$m = \frac{4\pi}{3} \cdot \varepsilon_0 V_{skin} a^3 \omega \tag{2}$$

where $V_{skin}$ is the potential of charges on the surface of the ball, V/m; $\omega$ is the frequency of the eddy current, Hz.

The electric field of a spherical $Fe_3O_4$ particle is considered a uniformly charged ball. Thus, the electric field on the surface of the shell of Al nanoparticles is determined by the expression [23,24]:

$$E = \frac{a^3}{3\varepsilon_0} \cdot \frac{1}{(a+d)^3} \tag{3}$$

where $\varepsilon_0$ is the vacuum permittivity, F/m. With an increase in the frequency of the external electromagnetic field of the microwave range, the magnetic conductivity and magnetic permeability of the material create a phase shift between the magnetization of the particles and the external alternating magnetic field.

The magnetic moment based on eddy currents is characterized by the frequency of external electromagnetic radiation and the shell radius. The energy associated with eddy currents affects the magnitude of the magnetic moment of the particle array and thus provides attenuation of the external electromagnetic field when an electromagnetic wave penetrates through the thickness of the material. External electromagnetic attenuation value of them can be determined from the equation of the skin layer thickness [23,24]:

$$\delta_{skin} = (\frac{\rho}{2\pi\omega\mu})^{1/2} \tag{4}$$

where $\rho$ is the resistivity, ohm·m; $\mu$ is the relative magnetic permeability.

According to the hypothesis, if the shell electrical conductivity value, which depends on Al nanoparticles amount on the surface of a $Fe_3O_4$ particles and interparticle adhesion, is characterized as a conductor, then this ensures a stable circulation of eddy currents in the shell of Al particles. For such case, the total magnetic moment of the $Fe_3O_4(Al)$ composite increases, and the EM reflection coefficient decreases.

If the electrical conductivity of the Al nanoparticle layer characterized as a dielectric, then the surface potential of the electric field strength of the dipoles of the spherical $Fe_3O_4$ particle increases. Consequently, the vector of the external electric field weakened, and as a result, the EM absorption coefficient decreases.

Figure 7 shows the calculated reflection loss value for the FC and FAC composites obtained under the suggested technology. In general, the $RL_{min}$ value less than $-10$ dB is comparable to attenuate 90% of incident electromagnetic wave, and thus, the $RL_{min} < -10$ dB has been regarded as an ideal microwave absorption material. It can be observed that the RL increases with a frequency range for FC and FAC samples. Sample FAC has superior

microwave absorption properties in both the $RL_{min}$ value and frequency width as compared with FC. An optimal $RL_{min}$ of −16.2 dB is obtained at 12.0 GHz for FAC sample. From Figure 7, we can conclude that FAC has a better microwave absorption property at identical coating thickness.

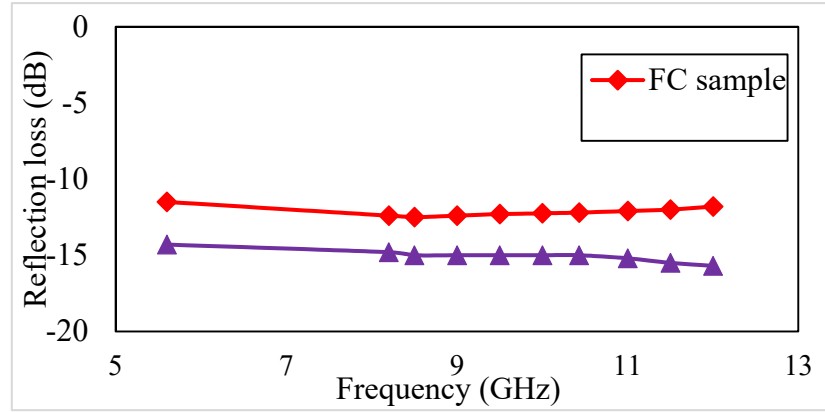

**Figure 7.** Reflection loss curves of the FAC and FC samples.

In comparison with known literature composite materials from Fe particles with different morphology, the RL values show stronger absorption characteristics [21] in the frequency range from 8 to 12 GHz. Figure 7 shows, that due to the Al shell, RL values of the FAC sample decreases without any resonance. This might be due to the wavelength increases in material because of dielectric permittivity decreases.

The value of the attenuation coefficient indicates that Al shell, as a result of $Fe_3O_4$ spherical particle coating by RMF method has Foucault currents. Thus, the hypothesis proposed in this paper is confirmed based on the results of the EM reflection spectrum characteristics. Further research will be focused on the analysis of magnetic properties of composite material obtained by RMF method, especially on magnetic anisotropy.

To investigate the probable mechanism for electromagnetic absorption performance, the electromagnetic parameters of FC and FAC composites are shown in Figure 8.

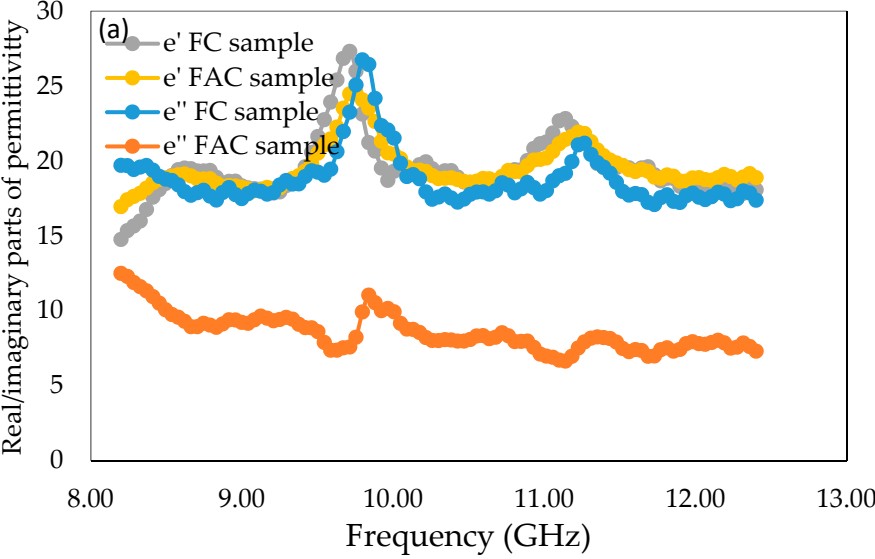

**Figure 8.** *Conts.*

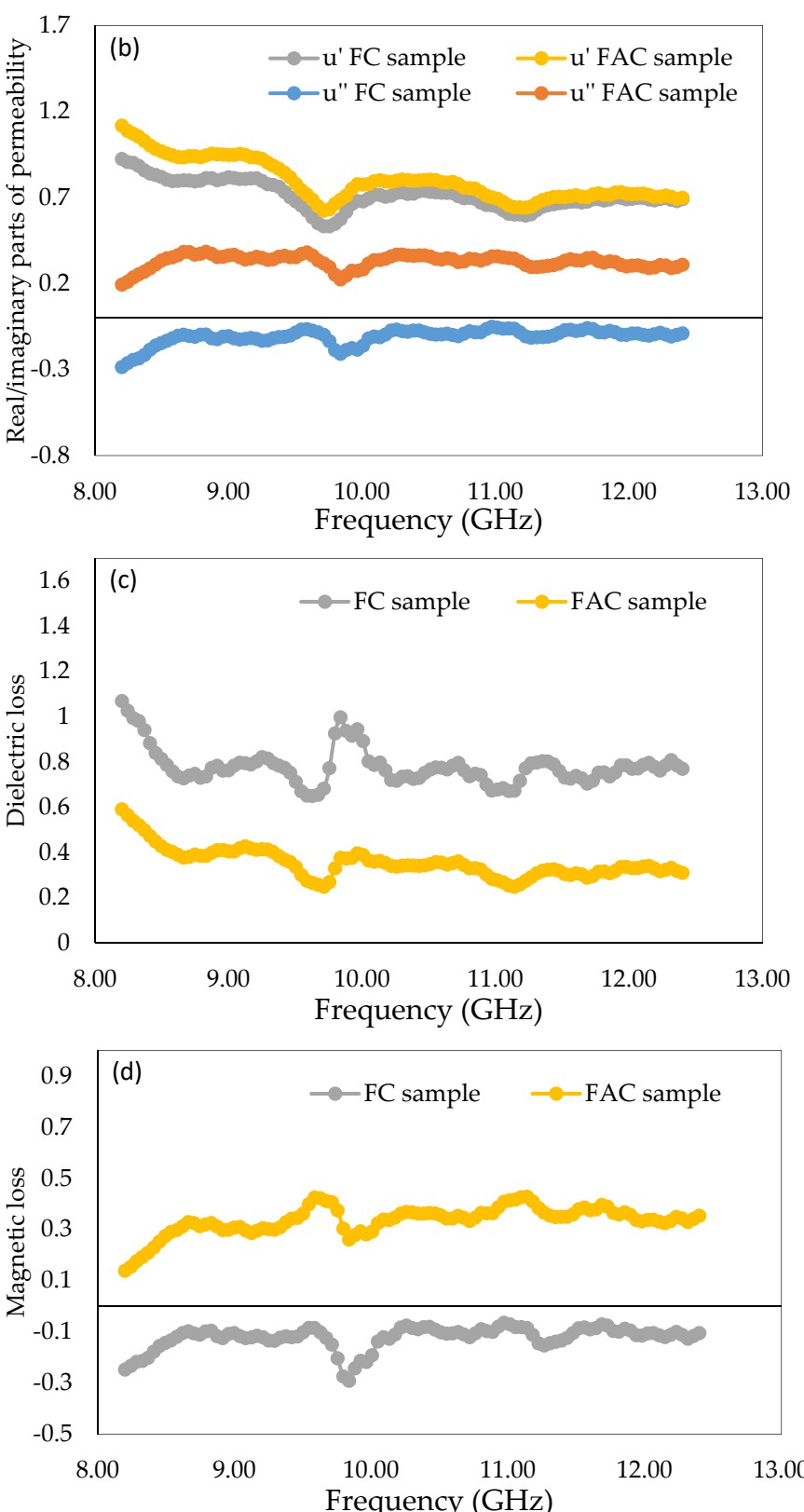

**Figure 8.** Electromagnetic parameters of FC and FAC samples: (**a**) real/imaginary parts of permittivity, (**b**) real/imaginary parts of permeability, (**c**) dielectric loss; (**d**) magnetic loss.

An absorbent with excellent microwave absorption may rely on the high impedance matching behaviour, meaning the absorbent can let more and more electromagnetic wave incidence in it. For high impedance matching properties, the material should meet the needs of the complex permeability and permittivity equally. Meanwhile, a big imaginary part of $\varepsilon''$ and $\mu''$ value is also quite important and represents the attenuation ability of the electromagnetic wave. Figure 8a shows the real/imaginary permittivity value; FAC exhibits a relatively lower $\varepsilon''$ value (8−13) and the same with the FC $\varepsilon''$ value (18−27). FAC has a better magnetic loss ability; both the real and imaginary parts of the permeability value are all slightly larger than those of FC (Figure 8b) due to the Al shell. From Figure 8c,d, we can conclude that the FAC has the stronger magnetic loss behaviour and weaker dielectric loss ability.

## 4. Conclusions

In this study, we designed novel porous core-shell $Fe_3O_4$(Al) structures with dense packing using the rotational magnetic field method. The mechanism of $Fe_3O_4$ particles coating by aluminium nanoparticles is described. The hypothesis put forward in this paper, about the effect of the surface charge density in the Al shell on the magnetization of the $Fe_3O_4$ core, is tested based on the EM reflection spectra. The calculated reflection loss implies that the porous core-shell $Fe_3O_4$(Al) composite materials present attractive electromagnetic absorption properties. The obtained characteristics of the reflection coefficient confirm suggested hypothesis for the composite material of $Fe_3O_4$(Al) particles.

The effective frequency bandwidth is up to 12 GHz (8−12 GHz) with a composite material thickness of 5 mm. Furthermore, the optimal frequency value for porous $Fe_3O_4$(Al) samples can reach −16.2 dB, higher than the −12.5 dB for the composite materials based on $Fe_3O_4$ particles only.

The improved microwave absorption properties may come from the high impedance matching behaviour and large $\varepsilon''$. Meanwhile, the $Fe_3O_4$(Al) structure with a dense packaging benefits electromagnetic wave scatters and results in enhanced microwave absorption properties.

## 5. Patents

Patent # 2544695. Dynamic filtering device. Shorstkii I.A., 2013.

**Supplementary Materials:** The following are available online at https://www.mdpi.com/article/10.3390/coatings11060621/s1, Video S1, Video S2.

**Author Contributions:** Conceptualization, I.S.; methodology, I.S.; writing—original draft preparation, I.S. and M.S.; supervision, I.S.; data curation, M.S.; funding acquisition, M.S. All authors have read and agreed to the published version of the manuscript.

**Funding:** The innovation project was carried out with the financial support of the Kuban Science Foundation in the framework of the Commercilizable scientific and innovation projects No. НИП-20.1-63/20.

**Data Availability Statement:** Data is available on request from the corresponding author.

**Acknowledgments:** Authors thanks center for collective use (CCU) "Research Center of Food and Chemical Technologies" for partially provided equipment.

**Conflicts of Interest:** The authors declare no conflict of interest.

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
