# Peer review of "Microwave Absorption Properties of Fe3O4 Particles Coated with Al via Rotating Magnetic Field Method"

_coatings, doi:10.3390/coatings11060621_

Round 1

Reviewer 1 Report

The authors reported their work on microwave absorption properties of Fe particles coated with Al. The work is useful, but English, spelling, and presentation need work.  For example, I noticed grammar errors in the second sentence in introduction “Actual problems of serious electromagnetic (EM) interference pollution arising from the rapidly expanding business of communication devices, such as mobile telephones, local area network systems and radar systems, has attracted great interest of scientists ……” 

Introduction

The authors stated that “In this study, we designed a novel porous core-shell Fe(Al) structures with a hexagonal particles packing as a new type of microwave absorption materials using rotational magnetic field method.”  Researchers have used magnetic field during the fabrication to influence the structure and packing of the materials. Can authors comment on what is the advantage of using rotational magnetic field method compared to others? Here are some papers authors can compare with.

Xiaowei Zuo ,Lin Zhang and Engang Wang, Influence of External Static Magnetic Fields on Properties of Metallic Functional Materials

Crystals 2017, 7(12), 374; https://doi.org/10.3390/cryst7120374

V.A.MilyutinI.V.Gervasyeva,  Journal of Magnetism and Magnetic Materials,  Volume 492, 15 December 2019, 165654, Journal of Magnetism and Magnetic Materials, Thermally activated transformations in alloys with different type of magnetic ordering under high magnetic field

J Zhao, J Yu, K Han, H Zhong, R Li, Q Zhai , Effect of Coil Configuration Design on Al Solidified Structure Refinement

Metals 2020, 10(1), 153; https://doi.org/10.3390/met10010153, 2020

Results and discussion:

Authors stated “Derived from the XRD patterns analysis of the FAC sample, it is found from Figure 2 that these strong diffraction peaks at 30,1°, 35,40°, 43,0°, 56,9° and 62.6°can be easily 107 ascribed to the crystal planes (220), (311), (400), (511), (440) ”  Are those peaks from iron or iron oxides? Please clarify in both the text and figure caption

line 113: Please correct grammar errors and carefully check other similar errors in the manuscript.  

Line 117: Please indicates how authors conclude the Al thickness was 200 nm.

Authors stated “Due to this complex motion, the nanoparticles were deposited evenly over the entire surface on the surface structure of the Fe particles .” The image appears to show that there are regions without any Al particle.  Can authors provide quantitarive estimation of uniformity of the deposition?

Line 129 “It can be seen that 129 the FAC sample presents porous core-shell Fe(Al) structures with a hexagonal particles packing.”  I cannot see what authors claimed in figure 4.

Line 142-143: Please correct English and carefully check other similar problems in the manuscript.  

Line 145: what is the meaning of “amd”?  Please check similar spelling errors.

How can authors assume that “For that, we put the assumption that there is a continuous aluminum layer on the Fe particles surface”?

Line 230: “In this study, we designed a novel porous core-shell Fe(Al) structures with a hexagonal particles packing using rotational magnetic field.”  I didn’t see any solid evidence of hexagonal packing.  Also, authors have a grammar error in the sentence.

Author Response

Dear Reviewer,

Thank you for the valuable comments intended for the improvement of the manuscript.

Response is attached,

Best regards

Reviewer 2 Report

The manuscript explains an unconventional method of preparing Fe/Al soft magnetic particles which can be used for EM shielding and microwave absorption in devices. The English should be revised by the authors to avoid some minor typo and mistakes in the manuscript. I have few comments for the author regarding the methods and structure of the Fe/Al particles which the authors should revisit before publication of the article:

1) XRD in Figure 2 has many peaks not indexed to Fe or Al. What are those phases? Al and Fe oxidize easily, and since Al are of nm scale oxidation is inevitable. How the authors are preventing oxidations in their process?

2) The authors claim that a core/shell form of particles are formed by their methods. However, the SEM image (Figure 4) does not show clearly any core/shell. If the authors highlight the core and shell in that image it would be helpful to understand.

Author Response

Dear Reviewer,

Thank you for the valuable comments intended for the improvement of the manuscript.

The response is attached,

Best regards

Reviewer 3 Report

The paper treats the fabrication of soft magnetic composites of Fe micrometer-sized particles that are coated with Al nanoparticles and their performance by microwave absorption and reflections. The authors claim that the Al nanoparticle coating on the Fe particles is uniformly distributed and show a stronger magnetic loss behavior as well as a weaker dieletric loss ability and better reflection characteristics.

The paper demonstrates a new way of the fabrication of uniformly coated Fe particles that show enhanced absorption properties in the microwave regime that may be interesting for future applications of composite materials. It contains a detailed description of the fabrication process and shows relevant experimental results concerning the resulting microstructure of the composite material as well as the microwave absorption properties that are improved in respect to other more common Fe particle composites. However, there are many major and also minor points that are needed to be addressed before a publication can be granted and many small details that need to be worked on.

  1. In a general view, the paper seems to be written in a hurry without the required elaborateness. Many misspellings, wrong declaration of figure numbers and missing articles or grammar mistakes are omnipresent in the text that makes the smooth reading very difficult. It must be revised before resubmission.
  2. Line 99: Even though one can look it up, for the general reader, the S parameter needs to be explained in general and what the meaning of the individual parameters (reflections) is.
  3. The XRD pattern suffers from quite bad statistics, particularly concerning the Al peaks. Why one has not measured longer. The Peak at 46.2 deg seems to be only one point which could be also a stochastically occurred intensity.
  4. The SEM pictures are not well presented. One needs to look at the small scale on their bottom to see the scales. The description should be more unified (e.g. Fig. 3 for (c) the magnification is given, for the others not). Moreover, from the series in Fig. 3, picture 3(a) shows the Fe particles before the coating and in 3(b) after the coating with Al nanoparticles. Why only one single particle is here present and around it, it seems to be somehow waste. This looks not as the well coated and distribution as it is indicated in the text. What the yellow dashed lined box means in 3(a) as it is seems not to be magnified picture in 3(b).
  5. Figure (4): the same question with the yellow dashed lined box arises as it does not seem to represent the magnified picture on the right. Moreover, in the text some of the conclusions needs to be better explained since it seems not so obvious from the presented SEM pictures: (a) the porous core shell structure has a hexagonal particle packing: one should somehow indicate it in the SEM pictures, as it is not clear; (b) how one can conclude that the pores should be filled with Al particles?
  6. The derivation of resulting electric field due to the Eddy current and the external electromagnetic attenuation is schematically explained and the details are referred to a reference that obviously only can be read if one can read in Russian what for an international publication is hardly practicable. One option is to add an appendix where the calculations are shown in (more) detail (recommended) or it needs to be added more details in the text. Or one can provide another meaniful reference (in English) that fills the gap.
  7. Line 192/199: Here not Figure 6 is meant, I assume Figure 5!
  8. In line 57, the common Fe(Al) composite material (FC) is introduced, but no specifications or reference is given. That makes it hard/impossible to judge about the comparison between FC and FAC. Here, some more details and references need to be provided or highlighted.
  9. Line 205: the conclusion from Figure 5 that the result of the cladding the Fe particles have Foucault currents needs a better justification or explanation and its consequences.
  10. In all graphs of Figure 6, the x-coordinate is in the wrong units (10^9 – 10^10 GHz is a little too much?)
  11. Line 226: wrong Figure denomination: Figure 9b -> Figure 6b.
  12. The discussion of Figure 6 is very superficial. The partly strong deviations of the FAC behavior in respect to the FC is not really discussed, however, here it would be interesting what the authors think the origins are and why.

Author Response

(The authors gave the same response as above.)

Round 2

Reviewer 1 Report

The authors revised their manuscript on microwave absorption properties of Fe oxide particles coated with Al. The manuscript is much clearer.  I still have a few suggestions. The first one is minor grammar errors. Compared to last version, these errors won’t cause significant confusion anymore, but it would be nice to correct them. Here is a typical example “Actual problems of serious electromagnetic (EM) interference pollution arising from the rapidly expanding business of communication devices, such as mobile telephones, local area network systems and radar systems, has attracted great interest of scientists in synthesis new types of microwave absorption materials with specific morphology, wide frequency range and strong absorption [2].” 

Here is another example “Our hypothesis based on the influence of the surface charges in the shell on the change in the magnetization of Fe3O4 particles.”

Introduction

The authors stated that “In this study, we designed a novel porous core-shell Fe(Al) structures with a hexagonal particles packing as a new type of microwave absorption materials using rotational magnetic field method.”  Researchers have used magnetic field during the fabrication to influence the structure and packing of the materials. Can authors comment on what is the advantage of using rotational magnetic field method compared to others? Here are some papers authors can compare with.

Revision 1 by authors “In comparison with static magnetic field (SMF) [16], strong magnetic field [17] and pulsed magneto-oscillation (PMO) [18] suggested method has few advantages: (1) an ability to control particles packaging density (from loose to dense structure), (2) possibility to provide core-coating process using dynamic magnetic platform.

Comment on revision 1:  In my opinion, both SMF and PMO can rotate effectively to provide similar magnetic fields that authors required, particularly a “dynamic magnetic platform”.  The major advantage appears to be more energy efficiency because the permanent magnet itself won’t use any energy.  The only energy the authors used was rotating the magnet.  

Results and discussion:

Authors stated “Derived from the XRD patterns analysis of the FAC sample, it is found from Figure 2 that these strong diffraction peaks at 30,1°, 35,40°, 43,0°, 56,9° and 62.6°can be easily ascribed to the crystal planes (220), (311), (400), (511), (440) ”  Are those peaks from iron or iron oxides? Please clarify in both the text and figure caption

Reply from the authors “In XRD patterns (fig 2) there was a mistake. The XRD patterns of Fe3O4 were swapped. XRD patterns were updated and clarified.”

Comment on revision 1:  I agree the revised version is much better. Can authors comment on if the final samples have any texture from XRD data. How will that affect the final properties?

I recommend that authors provide error bars in the graphs. In the manuscript, authors stated “…the optimal frequency value for porous Fe3O4(Al) samples can reach −16.2 dB, higher than the −12.5 dB…” Can authors provide error bars for both data so that readers can understand how significant the improvement is. If authors can do a statistical analysis, such as a t-test, it will be even more convincing.

Author Response

Please find my answers attached in a word file.

Reviewer 3 Report

The authors addressed all points and improved the manuscript to a degree that it can be accepted, howe after considering two modifications:

  1. Even though the English misspellings were removed and the language improved, the grammar is still in a bad shape. Often the articles are missing. It is absolutely recommended that the authors should get some advice of a native English speaker to correct for these or find some other solution to make the text more readable.
  2. It was definitely a good step to exchange some of the figures, in particular, the SEM pictures. One problem exists still for the yellow dashed-lined box of figure 5b where figure 5c is the according magnification. The pictures do not really match and checking for the scales, going from 10 micrometer to 100 nanometers, which is a magnification of 100 times, it cannot be correct considering the size of the cutout. That needs to be changed or better explained.

Author Response

(The authors gave the same response as above.)
